# Up-regulated chitinase-like protein-1 promotes tumour growth while physiological levels are protective

Shuangye Yan, Sam Holt, Xiao Gao, Joanna B Wilson

The human and mouse orthologues *CHI3L1*/*Chil1* encode a chitinase-related protein that is catalytically inactive. It is overexpressed in multiple inflammatory disorders and cancers and is thought to be pro-inflammatory and immune modulatory, but its role in tumorigenesis is unclear. Nevertheless, it has been proposed as a therapeutic target. To explore this, we have used Chil1 knockout compared with Chil1 WT mice, in combination with a transgenic mouse model (encoding the latent membrane protein-1 of EBV) of carcinoma-prone chronic skin inflammation. The data reveal that although high levels of Chil1 are pro-inflammatory, this can ameliorate the consequent tissue damage. Moreover, although high-level Chil1 promotes carcinoma growth, physiological levels inhibit the formation of papillomatous lesions and similarly inhibit the growth of tumours from transplanted cells. Furthermore, tumours arising in the Chil1 knockout mice showed reduced leukocyte infiltration, consistent with an impaired anti-tumour response. These dual roles warrant caution in developing and exploring therapeutic drugs that might abrogate Chil1 expression or effect.

## Introduction

Humans and mice, like other mammals, encode a family of proteins termed chitinase-like proteins or chilectins (CLPs or Chils), which have evolved from, and are similar in structure to, active chitinase enzymes, but are catalytically inactive (Hussain & Wilson, 2013). The human gene, *CHI3L1* (also known as *YK40*, *YKL40*, *GP39*), and the mouse orthologue, *Chi3l1* (also known as *Chil1*, *Brp39*, *Gp39*; hereafter referred to as *Chil1*), encode a secreted protein associated with inflammation, tissue repair, and remodelling. CHI3L1 is found at elevated levels in the pathological tissues of numerous chronic inflammatory diseases and a variety of autoimmune disorders, including rheumatoid arthritis, systemic lupus erythematosus, osteoarthritis, inflammatory bowel disease, multiple sclerosis, psoriasis, and Kawasaki disease (Vos et al, 2000; Vind et al, 2003; Comabella et al, 2010; Kim et al, 2017; Baran et al, 2018). Furthermore, the overexpression of CHI3L1 has been identified in multiple human tumours (in the tissue or in patient serum) and is associated with poor prognosis, including colorectal cancer, breast cancer, ovarian cancer, melanoma, prostate cancer, Hodgkin's lymphoma, glioblastoma, and nasopharyngeal carcinoma, amongst others (Cintin et al, 1999; Hogdall et al, 2003; Johansen et al, 2003; Brasso et al, 2006; Schmidt et al, 2006; Biggar et al, 2008; Shao et al, 2009; Li et al, 2024). Similarly, mouse Chil1 and the CLP homologues Chil3 (also known as YM1) and Chil4 (also known as YM2) have been found to be highly overexpressed in models of inflammation and cancer, including models of asthma and airway inflammation, osteomyelitis, breast carcinoma, and chronic skin inflammation and carcinoma (Webb et al, 2001; Zhao et al, 2007; Lee & Elias, 2010; Lee et al, 2011; Nikota et al, 2011; Qureshi et al, 2011; Chen et al, 2017). As such, CHI3L1 has been proposed as a diagnostic marker and possible risk factor for several diseases (Luo et al, 2017; Qiu & Zhang, 2022) and further as a therapeutic target (Lin et al, 2019; Jeon et al, 2021; Ma et al, 2021; Lee et al, 2022; Yu et al, 2022; Guetta-Terrier et al, 2023).

CHI3L1/Chil1 has been demonstrated to impact several signalling pathways, in different settings, including the MAPK/ERK, Jnk, Akt, and Wnt/$\beta$-catenin pathways (reviewed in Yu et al [2024]). One mechanism identified is through stimulated binding of the cytokine IL-13 to its receptor IL-13$\alpha$2 in macrophages, and another through association with CD44 leading to Myc-associated zinc finger protein (MAZ) activity in glioma stem cells (He et al, 2013; Kwak et al, 2019; Guetta-Terrier et al, 2023). However, the role of the increased expression of IL-13$\alpha$2 observed in some cancers is ambiguous, found to promote invasion and metastasis in pancreatic and ovarian cancer (Fujisawa et al, 2009, 2012), yet inhibit progression in pancreatic and breast tumours in another study (Kawakami et al, 2001). Furthermore, CHI3L1/Chil1 and the mouse homologues Chil3 and Chil4 have been found to impact the regulation and function of several immune cells, including macrophages and T cells (Huang et al, 2021; He & Kok, 2023; Heyndrickx et al, 2024; Yu et al, 2024).

Inflammation and the factors that control it are repeatedly revealed as double-edged swords, promoting healing and tissue

School of Molecular Biosciences, College of Medical, Veterinary and Life Sciences, University of Glasgow, Glasgow, UK

Correspondence: joanna.wilson@glasgow.ac.uk
Xiao Gao's present address is Office of Science and Technology Administration, Beijing Tiantan Hospital, Capital Medical University, Beijing, China

repair under some circumstances (especially with respect to the acute response to tissue injury or infection), while exacerbating pathology in other settings, particularly where inflammation is chronic. As such, the role inflammatory factors such as CHI3L1/Chil1 might play in inflammation-associated tumorigenesis is unclear.

In this study, we sought to clarify the contribution of Chil1 to chronic skin inflammation, carcinogenesis, and tumour progression using Chil1 knockout (KO) mice and an established transgenic mouse model of carcinoma-prone, chronic skin inflammation. Transgenic L2LMP1 mice display a phenotype of chronic skin inflammation with hyperplasia, most notably in hairless regions such as the ear pinnae. Multiple cytokines are deregulated in the affected tissue (Hannigan et al, 2011), which has been categorised into five progressive phenotypic stages (Stevenson et al, 2005). Briefly, these are as follows: stage 1: increased vascularisation; stage 2: hyperplasia and onset of inflammation; stage 3: extensive hyperplasia and chronic inflammation; stage 4: tissue degeneration; and stage 5: extensive degeneration. Occasional papillomatous lesions arise both around the head and ears and on the dorsal skin of the mouse. Treatment of the mice with the antioxidant N-acetylcysteine ameliorates the phenotype by reducing the inflammation (reducing the leukocyte infiltrate and reactive oxygen species in the affected tissue) and delaying its consequences (Gao et al, 2017). Considerable overexpression (as much as 20-fold) of Chil1, Chil3, and Chil4 is observed in the pathological tissues throughout the progression of this inflammatory phenotype and also observed in papillomas and carcinomas arising in the model (Qureshi et al, 2011). However, it is not clear whether this overexpression is causal or consequent to the pathology. This leads to the questions, are the Chils overexpressed as part of the tissue homeostatic mechanism to redress the persistent aberrant state? Alternatively, are they expressed as part of the chain of events initiated by the inflammatory oncogenic LMP1 transgene and contributing to the phenotype (Charalambous et al, 2007; Hannigan et al, 2007; Hannigan et al, 2011)? This study aimed to address these questions with respect to Chil1.

# Results

## Chil1 promotes inflammation but protects against damage

In order to assess the contribution of Chil1 to chronic inflammation, the L2LMP1 transgene was bred into a Chil1KO background to establish LMP1tg/Chil1KO mice. Transition between the progressive inflammatory stages shown by the ear pinnae (stages 1–5) was monitored (Figs 1A–C and S1). The average age of transition and average duration of each stage revealed that the absence of Chil1 (LMP1tg/Chil1KO), compared with WT Chil1 (LMP1tg), prolonged the phenotypic passage through stages 1 to 3 but accelerated passage from stage 4 to 5. The phenotypic stages 1–3 reflect direct inflammatory characteristics of increased vascularisation, inflammatory cell influx, rubor, and hyperplasia, whereas stages 4 and 5 reflect the destructive consequences of chronic inflammation including tissue erosion and necrosis. Therefore, Chil1 appears to contribute to LMP1-induced inflammation. However, the acceleration noted in the tissue erosion aspect of the

phenotype in LMP1tg/Chil1KO mice compared with LMP1tg (Chil1WT) mice suggests that Chil1 partially protects against the consequent tissue damage.

We have previously demonstrated that the cathepsin-activated fluorophore panC680 can be used to quantify the tissue inflammation in vivo, with the ear pinnae of LMP1tg mice showing significantly higher product radiance compared with WT mice (Gao et al, 2017). To quantify the impact of Chil1 upon the chronic inflammatory phenotype, mice were assessed by the in vivo imaging system (IVIS) at 100 d old (when mice typically show phenotypic stage 3) (Fig 1D). At all post-injection (pi) measurement points, mice of the LMP1tg/Chil1KO group showed significantly lower average radiance (corresponding to lower inflammation) compared with LMP1tg (Chil1WT) mice. These data support the observational assessment that the absence of Chil1 reduces the inflammation at this stage, indicating that the high level of Chil1 in the LMP1tg/WT tissue contributes to the LMP1-induced chronic inflammation.

Chil3 and Chil4 are rodent homologues of Chil1 (Hussain & Wilson, 2013), which along with Chil1 become highly induced in the inflamed LMP1tg tissue (Qureshi et al, 2011). Examination of protein levels in the ear pinna tissue showed that with knockout of Chil1, the induction of Chil3/4 was less marked in the KO background in older mice compared with Chil1WT, but is not higher at any age, demonstrating that there is no compensatory expression of Chil3/4 in the Chil1KO, instead suggesting there may be a feedforward action from Chil1 to Chil3/4, seen in older mice (Fig 2).

## Chil1 protects against further phenotypes

Occasional spontaneous small papillomas arise in the LMP1tg mice (Stevenson et al, 2005). The proportion of mice showing papillomas (in any location, e.g., head, dorsal skin) was significantly higher in the LMP1tg/Chil1KO group compared with LMP1tg, demonstrating that Chil1 protects against benign lesion formation in the LMP1tg mice (Fig 3A). A further phenotype was notable in the LMP1tg/Chil1KO mice affecting the eyes and area around the eyes. First observed as periocular inflammation, this could lead to closure of the eye with microphthalmia (Fig 3B–D), usually evident in one eye, sometimes in both eyes. Very occasionally observed in the LMP1tg mice and Chil1KO mice (and rarely in WT mice), the incidence of this phenotype was significantly higher in LMP1tg/Chil1KO, indicating that Chil1 protects against periocular inflammation and consequent microphthalmia induced by LMP1.

## Chil1 inhibits tumour formation, but promotes growth of established lesions with induced expression

Examination of the phenotype in these mice revealed contrasting actions of Chil1. Although Chil1 promotes the LMP1-induced inflammation, it appears to protect against benign papillomatous lesion formation and tissue damage, displayed by both the erosive dermatitis and microphthalmia observed in the LMP1tg/Chil1KO mice. To explore the role Chil1 might play in inflammation-associated tumorigenesis further, the mice were treated with topical chemical carcinogens (CC). We have previously reported that transgenic LMP1 expressed in the skin promotes CC-induced lesion formation, but limits lesion growth mediated via the tumour

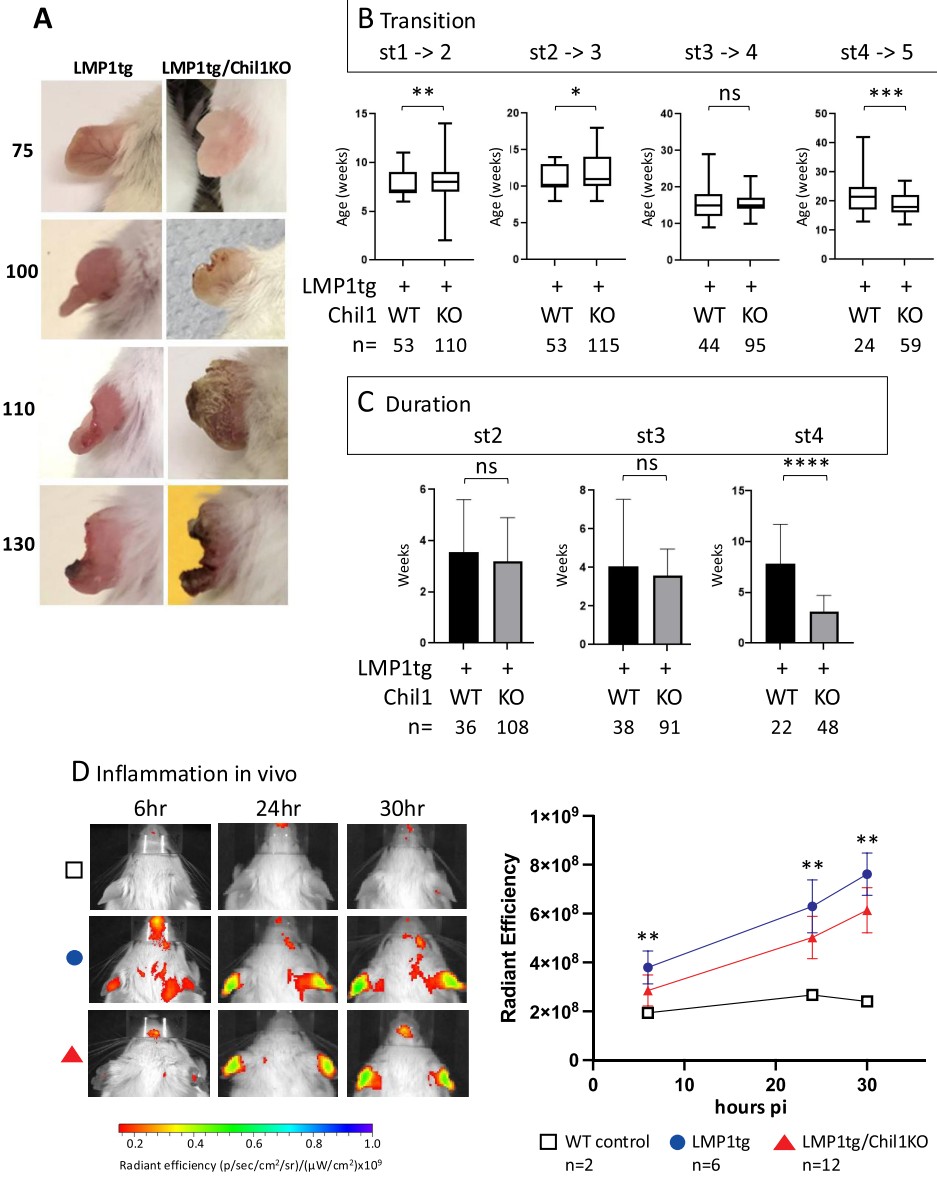

**Figure 1. Chil1 enhances inflammation but protects against damage.**
**(A)** Representative images of ear pinnae of mice in the LMP1tg and LMP1tg/Chil1KO groups, with age in days shown to the left. For the LMP1tg images: age 75/stage 2; age 100/stage 3; age 110/stage 3; age 130/stage 4. For the LMP1tg/Chil1KO images: age 75 and 100 show less rubor and tissue thickness; age 110 and 130 show more extensive necrosis and erosion. **(B, C)** LMP1tg and LMP1tg/Chil1KO mice were monitored for the age of transition between the defined ear pinna phenotypic stages (st) 1–5. Although progression of the phenotype is a continuum, distinct features have been used to characterise transition from one stage to the next (noted thickening [reflecting hyperplasia] defines transition from 1 to 2, first appearance of pinna edge erosion defines the transition to stage 3, greater than 50% edge erosion defines transition to 4, and extensive erosion and necrosis define transition to 5). **(B)** Plots show the average age of transition (1–2, 2–3, 3–4, 4–5 as indicated). **(C)** Graphs show the duration at each stage (2, 3, and 4 as indicated). **(D)** Inflammation was quantified in vivo by imaging mice aged 100 d old. Radiant efficiency $(p/sec/cm^2/sr)/(\mu W/cm^2)$ was measured at time points post-injection (pi: 6, 24, 30 h), ROI = each ear pinna. The left panel shows representative examples from each group. The right panel shows the average radiant time course (hours pi). Statistical significance was determined by a $t$ test between LMP1tg and LMP1tg/Chil1KO, where ns, not significant; *$0.05 \geq P > 0.01$; **$0.01 \geq P > 0.001$; ***$0.001 \geq P > 0.0001$; ****$P \leq 0.0001$. Error bars show the SD. Sample number (n) is indicated.

suppressor p16/Ink4a (expressed from the *Cdkn2a* locus) (Curran et al, 2001; Macdiarmid et al, 2003), which is known to induce senescence in response to oncogenic signals, thereby inhibiting neoplastic transformation (reviewed in [He and Sharpless [2017]]). Now using a distinct LMP1 transgenic line, despite the different transgene promoter and viral strain of LMP1, again LMP1 was found to promote CC-induced lesion formation, leading to earlier lesion appearance and significantly more lesions compared with WT mice, consistent with our earlier study and again noting that lesions grew larger in WT mice (Fig 4A, D, G, and J).

To assess the action of Chil1 in this context, lesion formation was compared between Chil1KO and WT backgrounds, using higher dose, but shorter duration TPA (6 wk [regimen 2] or 10 wk [regimen 3]). Lesion formation occurred earlier in LMP1tg mice compared with non-transgenic controls as observed with low-dose TPA

(regimen 1). Furthermore, lesion formation occurred earlier in LMP1tg/Chil1KO mice compared with LMP1tg (Fig 4B and C). Even with the exclusion of the pre-existing papillomas from the count, de novo lesion formation occurred significantly earlier in the LMP1tg/Chil1KO group (Figs S2 and S3). Limiting the TPA treatment to 6 wk resulted in some LMP1tg-negative (WT and Chil1KO) mice remaining lesion free for the duration of the study (Fig 4B).

Although of shorter treatment duration, increasing the TPA dose had two effects. First, the gap in total lesion numbers induced by LMP1 (LMP1tg) compared with WT was closed, such that by 10–11 wk (for both regimens 2 and 3), there was no significant difference in total lesion numbers (Fig 4E and F). Presumably, increasing the dose of the promoting chemical TPA obviated the promoting action of LMP1. However, the absence of Chil1 led to fewer lesions, most notably with treatment regimen 3 (Figs 4F and S4). This difference

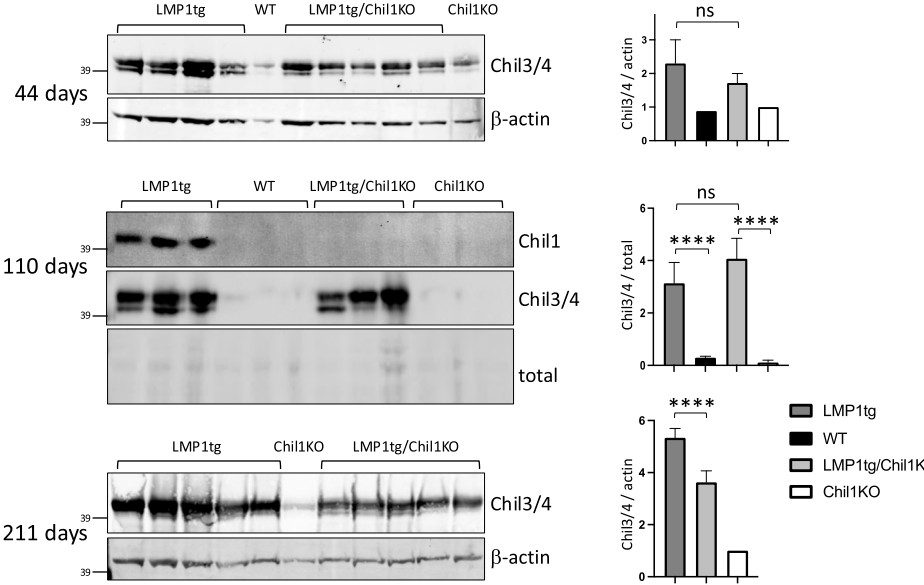

**Figure 2. Induction of Chil3/4 levels is damped in older Chil1KO mice.**
Protein extracts from ear pinna from mice aged 44, 110, and 211 d (as indicated; LMP1tg mice typically at stage 1, 3, and 5 respectively) were Western-blotted and probed with antibodies to Chil1, Chil3/4 (Chil4 running slightly faster than Chil3), and β-actin as indicated (39 kD marker is indicated) or total protein stained by Ponceau (as a loading control). Representative images from a range of ages are shown. Graphs show normalised band intensities of the blot data shown, taking Chil3 and Chil4 together (relative to Chil1KO mice set at 1). Statistical significance was determined by ANOVA (110 d), showing high significant difference for LMP1 and no significance for Chil1 across the 4 groups, or a t test examining the impact of Chil1 in the LMP1tg background (44 and 211 d), where ns, not significant; *0.05 ≥ P > 0.01; **0.01 ≥ P > 0.001; ***0.001 ≥ P > 0.0001; ****P ≤ 0.0001. Error bars show the SD.
Source data are available for this figure.

was also manifest in taking lesion size into account, with fewer larger lesions in the Chil1KO mice compared with WT (Figs 4H, I, K, and L, S5, and S6). Therefore, contrasting with the observation that Chil1 inhibits lesion formation (evidenced by more lesions in untreated Chil1KO mice and earlier lesion formation with CC in Chil1KO mice), these further data indicate that with continued CC treatment, Chil1 contributes to lesion formation and growth, in both LMP1tg mice and non-transgenic (WT) mice.

The second effect of a higher TPA dose was that lesion growth in WT mice was substantially accelerated, such that mice were necessarily removed from the study before the maximal endpoint. With regimen 2, 39% of the WT mice were removed from the study because of reaching the lesion load limit, with 50% of mice removed from the study (for all reasons) by week 20. Contrastingly, mice did not reach lesion load limit in the other three groups (Fig 5A–C). With regimen 3, 77% of the WT mice were removed from the study because of reaching the lesion load limit, with 100% of mice removed from the study by week 20 (Fig 5B and C). The proportion of mice in the other three groups reaching the lesion load limit was less than 20% by the end of the study (28 wk). The data show significant difference between WT and each of the other three groups. The difference in lesion growth rate between WT and LMP1tg is consistent with our earlier studies, which revealed that growth of lesions expressing the LMP1 oncogene was inhibited by *Cdkn2a* action (Macdiarmid et al, 2003). In the absence of Chil1, in LMP1tg/Chil1KO mice, the lesions were similarly slow-growing. Surprisingly, lesions in the Chil1KO mice (not harbouring the LMP1 transgene) were also slower growing than WT. However, no differences were observed in the time frame for conversion of papilloma to overt carcinoma (Fig S7).

To assess whether Chil1 levels might underlie these contrasting observations, expression was assessed in dorsal skin (the area subject to CC treatment) compared with ear pinnae and papillomatous lesions. Although Chil1 and Chil3/4 are induced in the epidermis of the inflamed ear tissue of LMP1tg mice, little or no induction was detected in the dorsal skin (Fig 5D). As previously

observed (Gao et al, 2017), IgH (~56 kD) becomes up-regulated in LMP1tg ear pinnae (compared with WT). Examination of CC-induced papillomas confirmed previous results (Qureshi et al, 2011), showing that Chil1 and Chil3/4 were up-regulated in the lesions from both LMP1tg and WT mice. In Chil1KO mice (both LMP1tg and non-transgenic), Chil3/4 was also induced in the lesions (Fig 5E and F). A hypothesis to explain these combined observations is that the physiological levels of Chil1 expressed in the untreated dorsal skin might be protective against lesion formation (evidenced by more lesions observed in LMP1tg/Chil1KO mice compared with LMP1tg/Chil1-WT), whereas the high levels of Chil1 observed in the tumours once formed (after CC treatment) may accelerate lesion growth (evidenced by reduced lesion growth in the CC-treated Chil1KO mice).

## Physiological levels of Chil1 inhibit tumour growth

To address this hypothesis, we sought to investigate the contribution of Chil1 to established tumours, especially in view of treatment modalities that might seek to inhibit Chil1 in cancer patients. To gain a comprehensive understanding of the impact of LMP1 and Chil1, we used several different cell lines, established from distinct CC-induced carcinomas, from each genetic group: LMP1tg+, LMP1tg/Chil1KO, Chil1KO, and WT mice (and of both sexes where possible). Although the CC-induced tumours show up-regulation of Chil3 and Chil4, and Chil1 (the latter only in Chil1WT), once established as cell lines in culture, Chil1 and Chil4 become undetectable and Chil3 is detected at very low level (Fig S8). Cells were transplanted by a single subcutaneous injection into recipients of the 4 genotypes and tumour development monitored. The time to tumour development differs for each cell line, consistent with its growth characteristics; however, for each cell line, tumours were collected from all mice in the cohort at the same time, to allow comparison. Sex of the cells and the recipients was also considered (Fig 6). Three LMP1tg cell lines (two male and

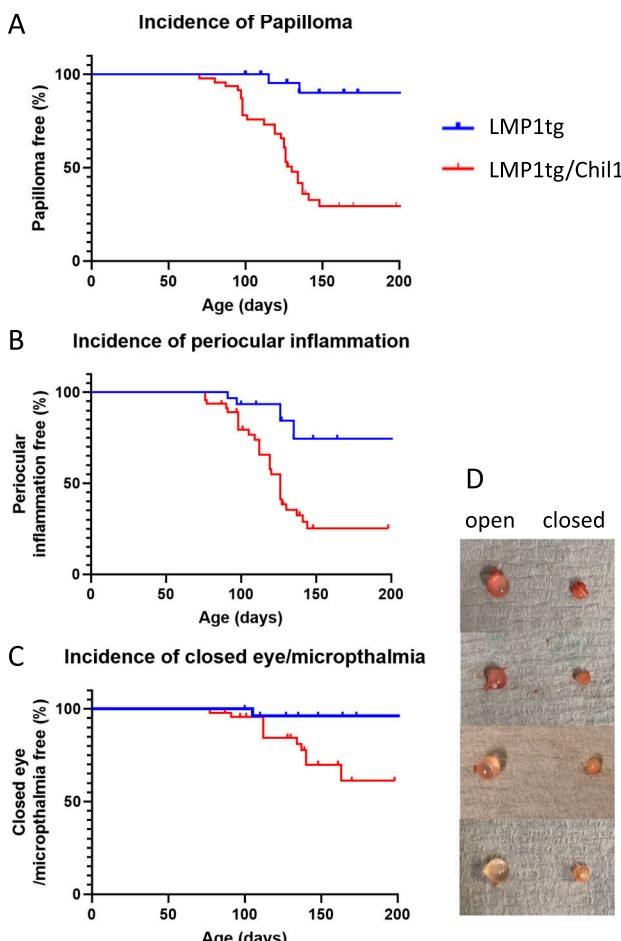

**Figure 3.  Chil1 protects against spontaneous papilloma formation and microphthalmia.**
**(A)** Papilloma formation at all sites was compared between LMP1tg (n = 30) and LMP1tg/Chil1KO (n = 47) mice, with the graph showing mice remaining lesion free over age. A log-rank test was employed showing significant difference at $P <$ 0.0001. **(B, C)** Eye phenotype for the same cohort of mice was categorised into periocular inflammation (B) and closed eye with microphthalmia (C), with mice remaining free of the phenotypes plotted against age. **(B, C)** A log-rank test was employed showing significant difference in both (B) $P < 0.0001$ and (C) $P = 0.012$. **(D)** Eyeballs from LMP1tg/Chil1KO mice showing microphthalmia (closed) were on average 49% the diameter of the unaffected eye (open) in the same individual and with cloudy appearance.

one female) showed clear tumour development in the LMP1tg (male) recipients (Fig 6A–C). Tumours in WT recipients (female) were significantly smaller or absent. The sex of the injected cell lines did not impact this result. No tumours formed in Chil1KO recipients (either LMP1tg-positive or LMP1tg-negative). With cell line 53.1576 (Fig 6C), this was explored further. No tumours developed in Chil1KO recipients by the 29-d time frame for this cell line, so Chil1KO recipients were followed out to 70 d post-injection, at which point still no tumours had developed.

Using a cell line that was WT (117.1451: LMP1tg-negative and Chil1WT) produced tumours in both LMP1tg and WT recipients, showing no significant difference between recipients carrying the LMP1 transgene or not (Fig 6D). However, using Chil1KO recipients with this cell line (both LMP1tg-positive and LMP1tg-negative)

produced significantly smaller or no tumours compared with Chil1WT recipients (Fig 6D). To extend this observation, this WT cell line (117.1451) was transfected with a Chil1-expressing plasmid or an empty vector control 1 d before transplantation. Tumour weight showed the same trend as non-transfected cells, showing larger tumours in LMP1tg and WT mice compared with Chil1KO recipients (both LMP1tg-positive and LMP1tg-negative) (Fig S9). However, unlike untransfected cells, in the resultant tumours, Chil1 was readily detected in tumours arising in LMP1tg recipients, faintly detected in WT recipients, and not detected in Chil1KO recipients. Given that after transfection, the cell population comprises a mixture of successfully transfected cells and untransfected cells, these data suggest that Chil1-expressing cells may have been selected for in the Chil1-WT recipients and/or selected against in the Chil1KO recipients.

Regarding all four Chil1WT cell lines (Fig 6A–D), the data suggest that cells carrying the LMP1 transgene limit tumour growth in transgene-negative (WT) recipients, which may reflect immune rejection targeting LMP1, despite the very low levels of LMP1 typically observed in these cell lines (Hannigan & Wilson, 2010). The data do not support that a sex difference between recipients is responsible for the difference in tumour growth observed between LMP1tg/male and WT/female recipients (shown in Fig 6A–C) as this is not seen using WT cells (Figs 6D and S9). Two primary explanations can be put forward as to why these Chil1WT cells developed significantly smaller or no tumours in Chil1KO recipients. The first is that Chil1 in the cells (despite being below the level of detection used here) led to antigenic rejection of the cells in Chil1KO recipients. The second is that Chil1 presence is required in recipients to enable tumour development. To distinguish between these possibilities, three independent Chil1KO cell lines (two male LMP1tg-positive and one female LMP1tg-negative) were generated and used in transplantation (Fig 6E–G). All three cell lines developed tumours in all recipient groups; however, tumours forming in Chil1KO mice (both LMP1tg-positive and LMP1tg-negative) were significantly larger than those forming in Chil1WT mice. This demonstrates that Chil1 is not absolutely required in the recipients for tumour formation and that the absence of tumours in KO recipients using Chil1WT cells may have been due to the first explanation proposed, that of immunological Chil1 rejection.

Therefore, the female, transgene-negative, Chil1KO cell line (with neither LMP1 nor Chil1 to act as antigens, Fig 6E) has revealed the contribution of Chil1 in the tumour environment, in the recipients. With larger tumours forming in the Chil1KO mice, this demonstrates that Chil1 (at physiological levels in the dorsal skin) in the recipient mice acts to inhibit tumour growth. Leukocyte infiltration into these tumours was examined, showing that there was a greater proportion of CD45[+] leukocytes in the tumours of Chil1WT recipients compared with Chil1KO (Fig S10). This observation is consistent with the Chil1KO mice showing an impaired immune response to the tumour cell insult.

## Discussion

We have exploited a model of carcinoma-prone, chronic skin inflammation to investigate the role of Chil1 in tumorigenesis and

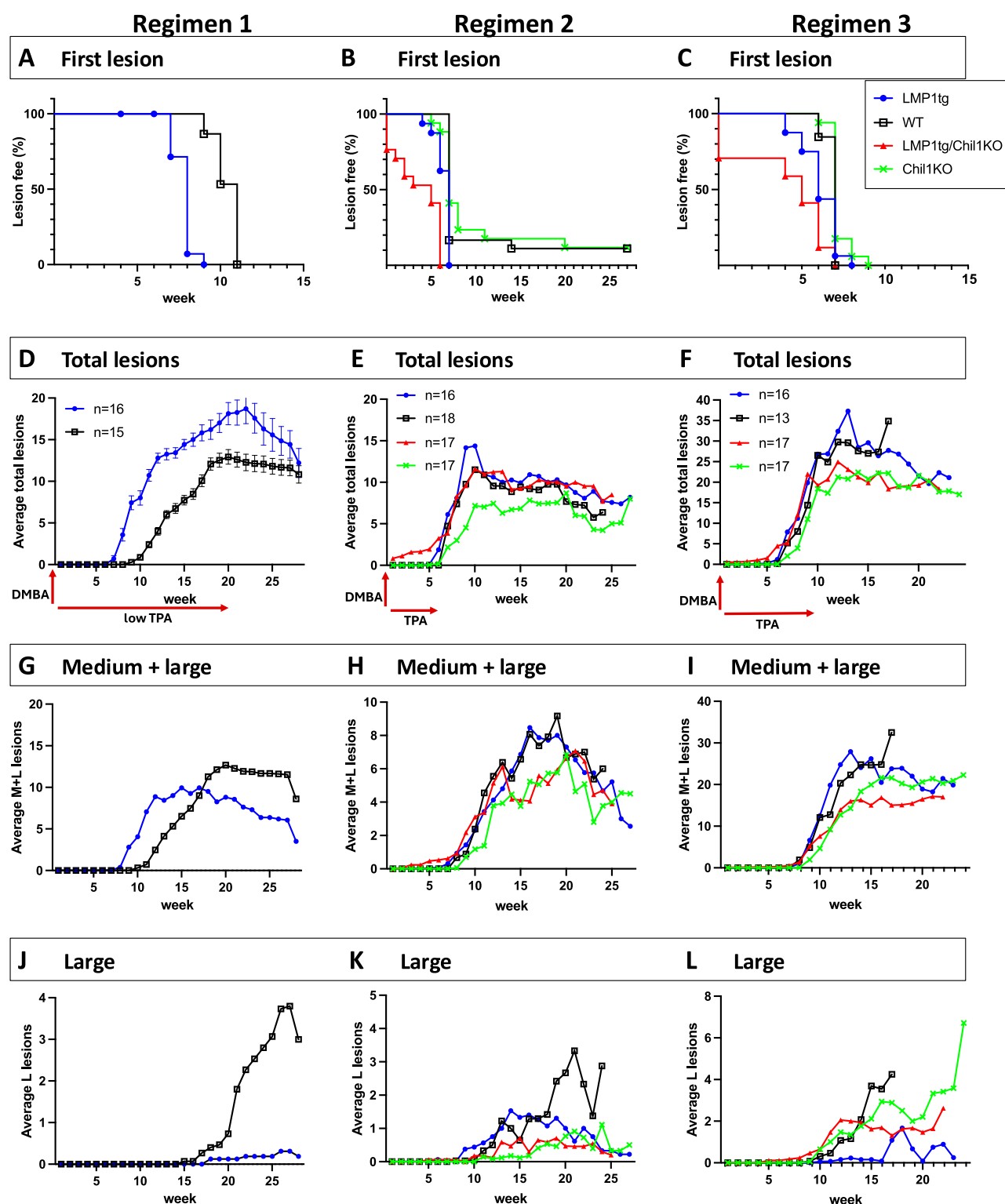

**Figure 4. LMP1 promotes lesion formation, while Chil1 protects against lesion formation but contributes to lesion growth.**
Mice were treated with CC following one of three regimens, as shown in the schematic (red arrows) under graphs (D, E, F). A single DMBA treatment was applied at week 0, followed by biweekly TPA treatments starting at week 1. Regimen 1: (left column) low-dose TPA, 20 wk. Regimens 2 and 3: (middle column and right column) medium-dose TPA, 6 or 10 wk, respectively. Number of mice (n) entered onto each of the three regimens (applying to all graphs for that regimen) is indicated in (D, E, F). **(A, B, C)** First lesion appearance is plotted showing the proportion of mice lesion free over time. Log-rank (Mantel–Cox) test was used to compare the curves, showing (A) significant difference between LMP1tg and WT, P < 0.0001. **(B, C)** Combined data up to week 7 for regimens 2 and 3 show significant difference between LMP1tg/Chil1KO and all other groups and between LMP1tg and all other groups, with no significant difference between WT and Chil1KO (Figs S2 and S3). **(D, E, F)** Average total lesions/mouse (each line

tumour growth. The LMP1 transgenic model studied here has been well characterised, showing that LMP1 induces skin inflammation and hyperplasia in proportion to its expression level (highest in hairless regions, notably the ear pinnae). The chronically inflamed tissue shows a substantial leukocyte infiltrate with a high level of intracellular reactive oxygen species and induced levels of numerous cytokines and inflammatory factors (Hannigan et al, 2011; Gao et al, 2017). In the ear pinna epidermis of these mice, Chil1, Chil3, and Chil4 are massively and consistently overexpressed (Qureshi et al, 2011). The inflammatory phenotype is apparent soon after birth and progressively worsens with age, with tissue erosion and necrosis. Conversely, no apparent induction of Chil1, Chil3, or Chil4 was observed in the dorsal skin, where little or no inflammation is observed (and transgene expression is low); however, occasional papillomatous lesions do appear on the dorsal skin and around the head in these mice (Stevenson et al, 2005). Cross-breeding of this line of transgenic mice to a Chil1 knockout background has revealed the role of Chil1, both in tissues where it is overexpressed in a WT background and where expression is at physiologically low levels.

Although it is currently unclear whether the mouse homologues Chil3 and Chil4 contribute to the observed phenotypes, the overexpression of both proteins was less marked in the chronically inflamed skin in the older Chil1 knockout mice, suggesting there may be feed-forward action from Chil1, to Chil3 and Chil4 induction.

Chil1 was found to be a pro-inflammatory factor in the LMP1-induced phenotype; in its absence, the inflammation was reduced. However, despite the abnormally high levels of Chil1 in the LMP1tg pathological tissues, it also exerts protective effects. In the LMP1tg/Chil1KO mouse inflamed tissues, degeneration was accelerated and an inflamed eye phenotype became apparent, indicating that Chil1 protects the inflamed tissue from erosion. This may reflect its anti-apoptotic and wound-healing activity, as seen in a model where Chil1 is reported to inhibit oxidant-induced lung injury (Sohn et al, 2010; Lee et al, 2011).

The role of Chil1 in tumorigenesis is more complex. The absence of Chil1 led to the formation of more spontaneous dorsal papillomas in the LMP1tg mice. In addition, the absence of Chil1 resulted in a faster response to CC-induced lesion formation in the LMP1tg mice. Furthermore, the absence of Chil1 enabled tumour growth from transplanted tumour cells.

Together, these data suggest that Chil1 acts to protect against LMP1-induced lesion formation and tumour growth. Conversely, the absence of Chil1 resulted in fewer total lesions with CC treatment, both in LMP1tg and in non-transgenic mice. Moreover, in the Chil1 knockout, lesion growth was slower, and this too was independent of the LMP1 transgene. Indeed, the rapid growth of CC-induced lesions in WT mice, compared with Chil1KO, was particularly marked. These data suggest that Chil1 is tumorigenic.

These apparently contradictory conclusions may be explained by Chil1 levels in the relevant tissue and reflect the importance of expression control and the dangers of chronic overexpression. In the dorsal skin, where the papillomatous lesions form and which is the location of tumour cell transplantation, Chil1 in the LMP1tg is not apparently higher in the skin. The low, physiological level of Chil1 may be providing protection against lesion formation and growth. Contrastingly, once lesions form with CC treatment, Chil1 levels are induced in the papillomas and carcinomas (both in LMP1tg and in WT mice) and it could be the abnormally high level that is pro-tumorigenic.

Human and mouse CHI3L1/Chil1 share 73% amino acid identity and are structurally highly similar. In humans, CHI3L1 expression is up-regulated in the tissues of numerous inflammatory disorders and multiple cancers, including the EBV-associated (LMP1-expressing) cancer nasopharyngeal carcinoma (NPC) (Zhao et al, 2020; Li et al, 2024). The overexpression of CHI3L1 in NPC cells is linked with tumour-associated inflammation and proliferation, and for this cancer, like several others, CHI3L1 is being considered not only as a diagnostic marker, but also as a potential therapeutic target.

In this study, we have explored the role Chil1 plays in a chronic inflammatory phenotype and consequent tissue damage. We have gone on to investigate the action of Chil1 in tumour initiation, tumour growth, and tumour support. A complex picture has emerged, with Chil1 acting to protect against tissue damage, whether expressed at low or high levels, yet its role in tumorigenesis and tumour growth apparently flips from protective at low, physiological levels to promoting at high levels. As such, these data emphasise that targeting CHI3L1 therapeutically must be approached with caution and in the knowledge of the full complexity of its action. With tumours that show high-level CHI3L1/Chil1 expression, its inhibition may well counteract its tumorigenic properties and be therapeutic; however, duration of such treatment would need to be carefully evaluated to avoid loss of any protection CHI3L1/Chil1 affords. For tumours with low or physiological levels of expression, anti-CHI3L1/Chil1 therapy has the potential to be deleterious.

# Materials and Methods

### Mice

The L2LMP1 transgenic mouse line 117 (L2LMP1.117) described previously, with the EBV-CAO strain of LMP1, was used in these studies (Stevenson et al, 2005). Herein, transgene-positive mice are denoted LMP1tg (or LMP1tg [Chil1WT] where relevant); transgene-negative sibling controls are indicated as WT. Note that the LMP1 transgene is integrated on the Y chromosome in this line, so all transgenic mice are male and negative sibling controls are female.

is plotted up to when 60% of mice were removed from the study). **(G, H, I)** Average medium and large lesions/mouse. **(J, K, L)** Average large lesions per mouse. **(D, G)** Under regimen 1 (D), *t* test from 8 wk of treatment shows significant difference at *P* < 0.001, from week 18 to week 22 shows significant difference at *P* < 0.02, and from week 24 shows non-significant difference (error bars show the SEM); (G) *t* test shows significant difference between weeks 9 and 15 and between weeks 23 and 29. **(J)** *t* test shows significant difference from week 20 reaching *P* < 0.00001. Statistical analysis of regimen 2 and 3 data is shown in Figs S3, S4, and S5.

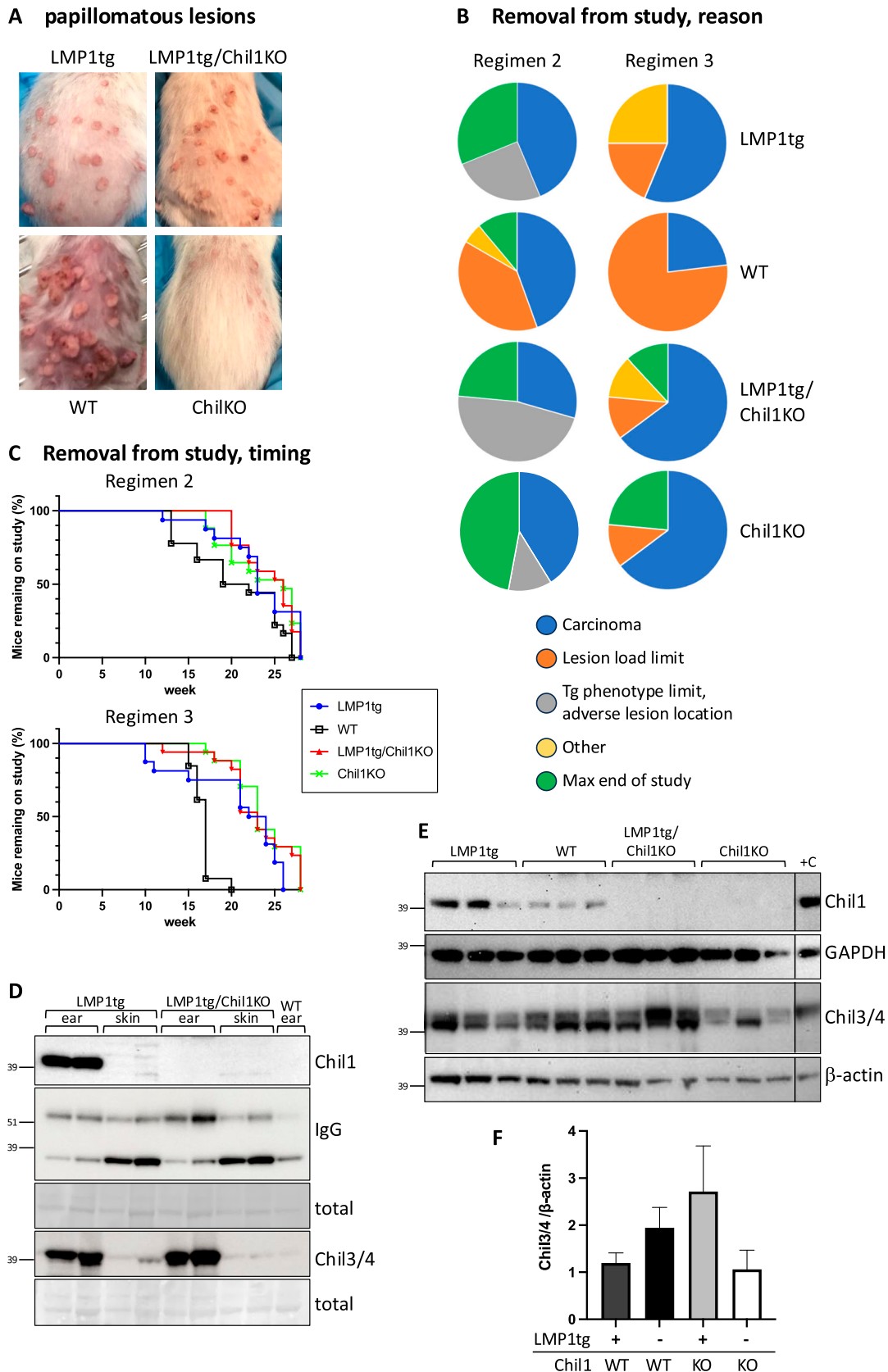

**A  papillomatous lesions**

LMP1tg    LMP1tg/Chil1KO

WT    ChilKO

**B  Removal from study, reason**

Regimen 2    Regimen 3

LMP1tg

WT

LMP1tg/
Chil1KO

Chil1KO

● Carcinoma

● Lesion load limit

● Tg phenotype limit,
adverse lesion location

● Other

● Max end of study

**C  Removal from study, timing**

Regimen 2

Regimen 3

- LMP1tg
- WT
- LMP1tg/Chil1KO
- Chil1KO

**D**

LMP1tg    LMP1tg/Chil1KO    WT
ear  skin    ear  skin    ear

Chil1
IgG
total
Chil3/4
total

**E**

LMP1tg    WT    LMP1tg/Chil1KO    Chil1KO    +C

Chil1
GAPDH
Chil3/4
β-actin

**F**

Chil3/4 /β-actin

LMP1tg  +  −  +  −
Chil1  WT  WT  KO  KO

Chil1 null mice, generated by deletion of the promoter and exons 1–6 (of 10) of the gene (a kind gift from Alison Humbles) (Lee et al, 2009), were crossbred with L2LMP1 mice, backcrossed four times to the FVB strain, then intercrossed, to establish the LMP1tg/Chil1KO line (93.75% FVB). LMP1 transgene-negatives in this line are denoted Chil1KO. The Chil1 genotype was determined from genomic DNA isolated from ear punch biopsy, by PCR using the primers: NeoF 5'TGCTCCTGCCGAGAAAGTATC, NeoR 5'CCAAGCTCTTCAGCAATACAC, Chil1F 5'GCCGGTCCAGAGGTCCTTGG, Chil1R 5'CCCAGGTCTGCACACCATAG. Phenotype scoring was conducted on a weekly basis. Mice were monitored at least twice weekly to assess health and well-being. Tissue samples were snap-frozen and stored at –80°C.

## Topical chemical carcinogen (CC) treatment

The chemical initiator 7,12-dimethylbenz[a]anthracene (DMBA) was delivered as a single dose at 25 $\mu$g (97.5 nMol, working concentration 125 $\mu$g/ml) at time point 0. Starting 1 wk after initiation (week 1), the chemical promoter 12-O-tetradecanoylphorbol-13-acetate (TPA) was used following one of three regimens: [1] (low) 3.125 $\mu$g (5 nMol) twice per week over 20 wk, [2] (mod) 6.25 $\mu$g (10 nMol, working concentration of 31.25 $\mu$g/ml) twice per week over 6 wk, or [3] (mod) 6.25 $\mu$g (10 nMol) twice per week over 10 wk. Note that TPA dose ranges between 2.125 $\mu$g (3.4 nMol) and 20 $\mu$g (32 nMol) have been used previously with FVB strain mice (Abel et al, 2009). Each treatment was delivered in 0.2 ml acetone on the dorsal skin (shaved at the start of the study and intermittently as needed). Mice were entered onto study at 8 wk old. The end of the study was maximally 40 wk post-first treatment with regimen 1 and 28 wk post-first treatment for regimens 2 and 3. Lesions were counted on all mice weekly and categorised into sizes (diameter): small: ≤1 mm; medium: >1 and ≤5 mm; large: >5 mm; and overt carcinoma. Mice were removed from the study because of [1] reaching the lesion accepted size limit (1.3 cm$^2$) (including overt carcinoma), or lesion ulceration; [2] reaching the total lesion load limit (determined by the number and size of papillomas); [3] any adverse lesion location, or transgenic phenotype limit (particularly affecting eyes and ear pinnae); and [4] any health issue not related to topical lesions.

## In vivo imaging system (IVIS)

Inflammation was quantified in vivo by intravenous injection of 0.5 nmol/mouse of an IVISense Pan Cathepsin 680 fluorescent probe (panC680) prepared in PBS. Epifluorescence was detected at $\lambda$ex 675 nm/$\lambda$em 720 nm at intervals 3–30 h post-injection, under isoflurane-induced anaesthesia. Radiance within the region of interest (ROI) was quantified using Living Image software, as previously described (Gao et al, 2017).

## Ethics statement

All animal work was conducted; the study and all protocols were approved, under UK Home Office licence and according to institutional, national (AWERB), and international guidelines under the UK's Animals (Scientific Procedures) Act 1986 Amendment Regulations 2013. This law is in line with the EU Directive on the Protection of Animals used for Scientific Purposes (Directive 86/609/EEC as updated by Directive 2010/63/EU).

## Data analysis

Statistical difference between groups was calculated using two-way ANOVA, $t$ test, or pairwise log-rank test as appropriate, using GraphPad Prism.

## Cell culture and transplantation

Mouse carcinoma cell lines were derived in culture from CC-induced carcinomas in mouse lines L2LMP1.117 or PyLMP1.53 as previously described (Curran et al, 2001) and were grown in DMEM with 10% FBS, 1% L-glutamine, 100 units penicillin/ml, 0.1 mg streptomycin/ml. Cell lines include the following: 53.278 (M/tg+/WT indicating male/LMP1tg-positive/Chil1WT), 117.1472 (M/tg+/WT), 117.1451 (F/tg-/WT), 53.1576 (F/tg+/WT), 117166.363B and 117166.363C (M/tg+/KO), and 117166.523 (F/tg-/KO). 10$^6$ cells were injected in 100 $\mu$l PBS dorsally and subcutaneously into recipients of each genotype (all FVB strain). When a tumour in a cell line cohort group reached the accepted size limit, the study was terminated for the whole group and tumours were collected and weighed. Cells transfected with plasmid were injected 1 d after transfection. Chil1 plasmid (pmChil1-his) encodes full-length mouse Chil1 cDNA with C-terminal his-tag; empty vector is pcDNA3.1.

## Western blotting

Proteins were extracted in RIPA buffer (150 mM NaCl, 50 mM Tris–HCl, pH 7, 1% Triton X-100, 0.1% SDS, protease and phosphatase inhibitors) and separated (20 $\mu$g per track) by SDS–PAGE

---

**Figure 5. Induced Chil1 promotes lesion growth.**
**(A)** Representative photographs of papillomatous lesions on mice from each group at week 9 of regimen 3. **(B)** Proportions (reasons as defined in the Materials and Methods section) of removal of mice from the study by the maximal endpoint, for each group. **(C)** Removal of mice from the study for all reasons is plotted against week of study for regimens 2 and 3. A log-rank (Mantel–Cox) test was used to compare curves, with WT showing significant difference from other groups (regimen 2: WT versus LMP1tg, ns; WT versus Chil1KO, $P$ = 0.03; WT versus LMP1tg/Chil1KO, $P$ = 0.03; regimen 3: WT versus LMP1tg, $P$ < 0.0014; WT versus Chil1KO, p=<0.0001; WT versus LMP1tg/Chil1KO, $P$ < 0.0001). **(D)** Chil1 and Chil3/4 are not detectably up-regulated in LMP1tg dorsal skin. Protein extracts from ear pinnae and dorsal skin (mice at 130 d old, LMP1tg ear stage 3) were Western-blotted and probed with antibodies to Chil1, Chil3/4, and IgG (IgH at ~56 kD, the lower band more prominent in dorsal skin than ear pinnae, is likely a form of IgL), with total protein by Ponceau stain indicated as a loading control. **(E)** Chil1/3/4 are up-regulated in CC-induced lesions. Protein extracts from medium to large papillomas after CC treatment were Western-blotted and probed with antibodies to Chil1, Chil3/4, GAPDH, or $\beta$-actin as indicated. Positive control (+C) extract from stage 5 LMP1tg ear tissue. **(D, E)** Protein ladder marker locations are indicated in (D, E). **(E, F)** Band intensity of Chil3/4 (together, normalised to actin from blots shown in (E)) is plotted; groups do not show statistical significant difference (note, for LMP1tg/Chil1KO versus Chil1KO, $P$ = 0.053).
Source data are available for this figure.

## LMP1tg/Chil1WT cells

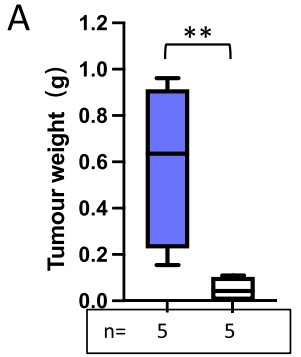

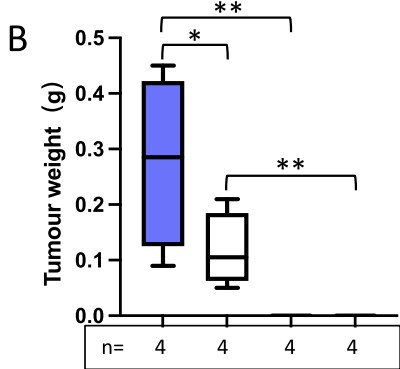

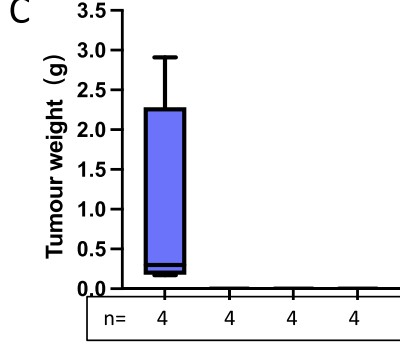

## WT cells

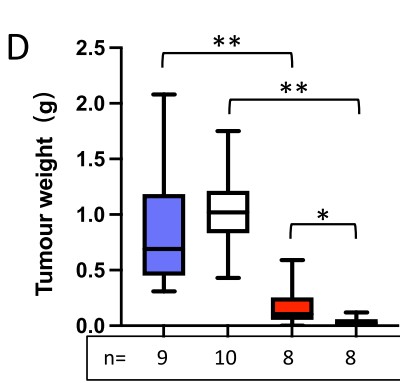

## LMP1tg/Chil1KO cells

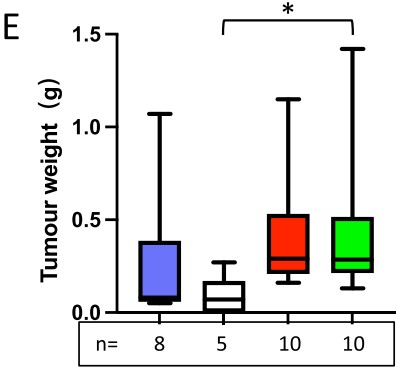

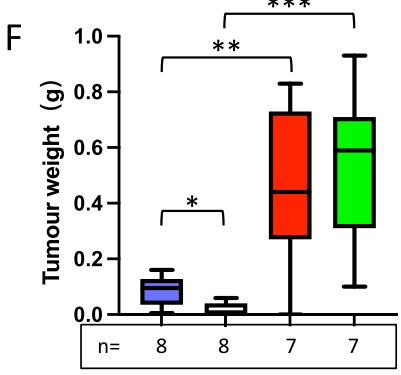

## Chil1KO cells

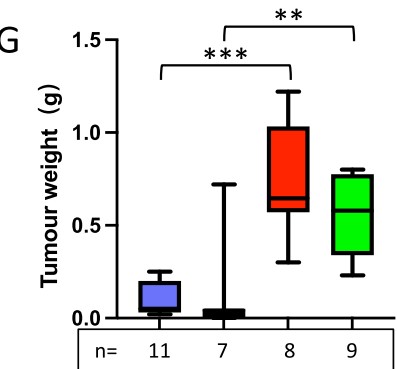

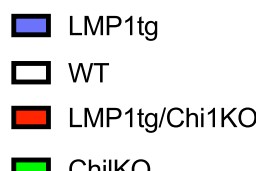

recipient genotypes

- LMP1tg
- WT
- LMP1tg/Chi1KO
- ChilKO

**Figure 6.   Impact of Chil1 upon tumour transplantation.**
Different cell lines, with differing LMP1tg and Chil1 status (as indicated in the headings), were injected subcutaneously into mice of recipient genotypes as shown (key). Tumours were allowed to form for a defined period, specific to each cell type, and were then collected and weighed. Box-and-whisker plots show resulting tumour weights. **(A)** 58.278 cells (M/tg+/Chil1WT), tumours collected at 21 d, (B) 117.1472 cells (M/tg+/Chil1WT), tumours collected at 26 d, (C) 53.1576 cells (F/tg+/Chil1WT), collected at 29 d, (D) 117.1451 cells (F/WT), collected at 14 d, (E) 117166.363C cells (M/tg+/Chil1KO), collected at 14 d, (F) 117166.363B (M/tg+/Chil1KO) collected at 16 d, and (G) 117166.523 (F/tg-/Chil1KO), collected at 14 d. **(A, B, C, D, E, F, G)** Tumour sizes were compared by two-way ANOVA, and, because of significant interaction (between the variables LMP1 status, Chil1 status), were assessed in pairs by a $t$ test: significance: *$0.05 \geq P > 0.01$; **$0.01 \geq P > 0.001$; ***$0.001 \geq P > 0.0001$; ****$P \leq 0.0001$ (otherwise not significant). **(C)** Note: part (C), significance testing is not possible as no tumours formed in three groups.

using precast NuPAGE/Bolt 4–12% gels (Invitrogen). SeeBlue Plus2 (Thermo Fisher Scientific) protein size markers were used. Blotting, washing, and antibody incubations (and blot stripping for reprobing) were performed as previously described (Hannigan et al, 2011). Antibodies (with dilutions) used were as follows: rat anti-Chil1 (R&D Systems), rabbit anti-Chil1 (Abcam), goat anti-Chil3/4 (R&D Systems), goat anti-actin (Santa Cruz), and mouse anti-GAPDH (all at 1:1,000); rabbit anti-Chil3/4(YM1/2) (Abcam) at 1:3,000; and rabbit anti-actin (Cell Signaling) at 1:5,000, followed by the appropriate anti-IgG HRP-conjugated secondary antibodies: goat anti-mouse, rabbit anti-mouse, goat anti-rabbit, goat anti-rat, or donkey anti-goat (1:5,000 or 1:10,000; Santa Cruz, Abcam, or Sigma-Aldrich); or Li-COR IRDye-conjugated secondary antibodies (1:3,000) donkey anti-goat and goat anti-rabbit. Detection of HRP conjugates was performed by enhanced chemiluminescence (ECL-Pierce, Thermo Fisher Scientific), and bands on blot images were quantified using ImageJ. Detection and quantification of Li-COR infrared fluorophores was by the Odyssey imager at the appropriate wavelength.

## Supplementary Information

## Acknowledgements

We thank several individuals who contributed to the data during their tenure as master's students, including the following: Rachel Neil, who contributed to IVIS experiments; Jiahua Niu who contributed to Fig 5E; and Heather Flanagan, who contributed to tumour cell transplant studies. S Yan was self-funded. X Gao was supported by a China Scholarship Council/University of Glasgow scholarship.

### Author Contributions

S Yan: data curation, formal analysis, funding acquisition, investigation, and methodology.
S Holt: investigation and methodology.
X Gao: data curation, formal analysis, funding acquisition, investigation, and methodology.
JB Wilson: conceptualisation, resources, data curation, software, formal analysis, supervision, funding acquisition, validation, investigation, methodology, project administration, and writing—original draft, review, and editing.

### Conflict of Interest Statement

The authors declare that they have no conflict of interest.

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
