## [Reviewer comments · Life Science Alliance]

Life Science Alliance

Upregulated chitinase-like protein1 promotes tumour growth while physiological levels are protective

Shuangye Yan, Sam Holt, Xiao Gao, and Joanna Wilson

DOI: <https://doi.org/10.26508/lsa.202403138>

Corresponding author(s): Joanna Wilson, University of Glasgow

Review Timeline:

Submission Date:	2024-11-18
Editorial Decision:	2025-01-10
Revision Received:	2025-06-10
Editorial Decision:	2025-07-09
Revision Received:	2025-07-28
Accepted:	2025-07-30

Scientific Editor: Tim Fessenden

Transaction Report:

January 10, 2025

Re: Life Science Alliance manuscript #LSA-2024-03138-T

Prof. Joanna Beatrice Wilson
University of Glasgow
School of Molecular Biosciences, College of Medical, Veterinary and Life Sciences
Glasgow G12 8QQ
United Kingdom

Dear Dr. Wilson,

Thank you for submitting your manuscript entitled "Upregulated chitinase-like protein-1 (CHI3L1/Chil1) promotes tumour growth while physiological levels are protective" to Life Science Alliance. The manuscript was assessed by expert reviewers, whose comments are appended to this letter. We invite you to submit a revised manuscript addressing the Reviewer comments.

Thank you for this interesting contribution to Life Science Alliance. We are looking forward to receiving your revised manuscript.

Sincerely,

B. MANUSCRIPT ORGANIZATION AND FORMATTING:

Reviewer #1 (Comments to the Authors (Required)):

This manuscript presents a thorough investigation of Chil1's roles in inflammation and tumor development using a variety of experimental approaches. The study's main strength lies in its comprehensive analysis of Chil1's functions in different contexts, revealing its complex and dual nature. The strengths of the studies are noted in many areas of manuscript: methodological approaches employing well-characterized transgenic mice (L2LMP1) and KO mice (Chil1^{-/-}) and multiple experimental approaches including phenotyping scoring, chemical carcinogenesis, and tumor cell transplantation. Most importantly, the studies provide a comprehensive feature of Chil1 function including dual role of Chil1 in inflammation and tumorigenesis. As a result, the studies nicely demonstrated dual regulatory role of Chil1- both proinflammatory (protumorigenic) and tissue protective effects on tissue destruction. Additional strength is identification of feed-forward relationship between Chil1 and its homologues Chil3 and Chil4. However, weaknesses are also recognized especially in the methodological approaches and results that did not provide for clear mechanistic understanding and explanation on the dual role of Chil1.

Followings are the specific comments that would increase the strengths of the manuscript.

1. The study relies heavily on mouse models, and the translational relevance to human cancers is not directly addressed or discussed.
2. The mechanisms by which Chil1 exerts its effects are not fully elucidated at the molecular level. The study does not delve deeply into the signaling pathways affected by Chil1, which could provide more mechanistic insight into its functions. Additional mechanistic studies exploring the differences in Chil1 signaling based on the level of Chil1 expression would be a significant help to address this issue.
3. The impact of Chil1 on specific immune cell population and their function is not thoroughly explored.
4. The potential interactions between Chil1 and other inflammatory mediators or oncogenic pathways are not extensively investigated.

Reviewer #2 (Comments to the Authors (Required)):

The manuscript by Yan et al aims to determine whether targeting CHI3L1 (Chil1) would be beneficial or detrimental in the setting of chronic skin inflammation and skin cancer, using Chi1-null mice and a transgenic mouse model of skin inflammation that can progress to cancer. Interestingly, lack of Chi1 ameliorated earlier stages of disease progression in the transgenic mouse model but exacerbated later stages. This suggested that normally Chil1 acts to promote inflammation but is protective against later tissue injury. Moreover, when specifically examining a carcinogen version of the model, Chil1 appears to protect against tumor initiation but promotes tumor growth. The authors hypothesize that these disparate effects are related to normal ('physiological') levels of Chil1 being protective, while the increased levels seen in tumors are tumor-promoting. To examine this idea, they created cell lines from tumors formed in the mice of different genotypes and then injected these into host mice of differing genotype. The results suggested that ablating Chil1 in the host is protective when the tumor cells have Chil1, but significantly worsens tumor phenotypes if the tumor cells lack Chil1. An explanation put forward by the authors is that Chil1 is itself antigenic thus leading to rejection in the Chil1-null mice of Chil1⁺ tumor cells. Additionally, total leukocyte recruitment into tumors appears higher in Chil1⁺ hosts, suggesting a potential immune-mediated restraint of tumor growth when Chil1 is present, although this is not further explored. Overall, the data would suggest that therapeutically targeting Chil1 in a tumor setting could lead to enhanced tumor growth.

This manuscript generally describes phenotypes rather than exploring mechanisms, other than showing a lack of compensatory Chil3/Chil4 expression in Chil1 KO conditions. This overall conclusion that therapeutically targeting Chil1 could be problematic is supported by the data and is valuable information. However, there are aspects of the manuscript that are difficult to follow and/or could be clarified.

For the injection experiments (Fig 6), the authors use mice of different sex as recipients, though cell lines represent both sexes. This makes interpretation difficult as it is known that anti-tumor immune response are different in males and females. By using transgenic males but WT females as recipients, the host responses could be as much influenced by the sex as by the presence of the transgene. This should at least be discussed as a variable.

The cell transplant experiments include several variables that complicate interpretation. By using cell lines established

separately from WT and KO mice, there are possibly multiple other differences amongst cell lines that could contribute to effects seen, rather than just the absence or presence of Chil1. Experiments with single cell lines (e.g Fig S10) are more straightforward. Again, this variable should be acknowledged and discussed.

The display of statistical significance is inconsistent from figure to figure. In figure 2, for the 100 day data, what genotypes are being compared for the significance analysis? In other places it is LMP1tg vs LMP1tg/Chil1 KO and WT vs Chil1KO, however the graphic here suggests that the comparison is LMP1tg vs WT and LMP1tg/Chil1KO vs Chil1KO. It is unclear why the 44 and 211 day data are analyzed pair-wise using a t-test, while the 100 day data is analyzed using ANOVA. In figures 3 and 4, statistical significance should be displayed on the graphs as done in previous figures.

Reviewer #3 (Comments to the Authors (Required)):

Chitinase-like proteins (CLPs) are highly expressed in many different diseases and cancers, with roles in regulating immune responses, tissue repair and fibrosis suggested. However, few studies have directly investigated the role of these proteins in tumourigenesis. This study aimed to investigate whether mice deficient in Chil1, one of these chitinase-like proteins, had altered prevalence and severity of carcinoma skin lesions using a transgenic mouse model. Data presented by the authors suggest that overexpression of Chil1 can promote development tumour growth, whilst physiological levels can prevent growth of transplanted tumour cells, demonstrating the complexity of CLP effects (e.g. pathogenic expression vs homeostatic expression)

This manuscript offers new evidence of the potential roles of Chil1 in tumour growth and highlights the complexity of targeting these proteins therapeutically to treat cancer. Overall experiments appear well designed, appropriately powered and statistically analysed. However, at times the data were a little confusing and inherently complex and perhaps some restructuring and using some schematics in the figures may better help to clarify these findings.

Specific comments

1. General: There are several grammatical errors in the figures and text that need to be fixed. Also at times the results section bordered on having too introduction and background information. Better citation to figures within the manuscript are needed. E.g description of figure 1 in the text doesn't refer to fig 1 a or b, and other figures e.g. fig 5e or 5f aren't referred to all in the text.
2. Not being familiar with the transgenic model system used throughout, I actually think it would be beneficial to show sup fig 1 within the main figures, perhaps incorporated into figure 1 to explain the transition between stages of inflammation/necrosis. Also, by not specifically referring to fig 1b, the description became confusing in relation to transition and duration.
3. Figure 2 shows expression of CLPs in the ear tissue at various stages (days) normalised to beta-actin levels. However, the middle panel - 100 days doesn't appear to show results normalised to beta-actin, so how can the authors be sure equivalent protein loading across samples and hence of their conclusions? Also are all the quantified data shown in the blots or are the blots representative. Details of n numbers etc need to be included in the figure legend. Chil3/4 is shown as a double band, does this relate to one protein being larger than the other? In some instances, it looks like the lower band is expressed at a lower level compared to the higher band?
4. At day 211, it appears that Chil3/4 expression is significantly reduced in Chil1KO mice. Considering these results, could the effects observed at later time points during stage 5, indicate that Chil3 and Chil4 in addition to Chil1 also may play a role in any outcomes?
It wasn't clear to this reviewer what mouse the eyeball images were from. Was this comparing transgenic mice with and without Chil1? How does this data support the claims that Chil1 protects against microphthalmia?
5. Figure 4 and 6 represents complex experimental design to answer specific questions, particularly for people not familiar with the transgenic model. I think it would be worth the authors adding a schematic at the start of the figure to show the different treatments between regiments etc more clearly. Also statistics on each graph in figure 4 would be useful. Figure 6 is particularly challenging given the different model system used, and readers could also miss the key points without a clearer explanation.
6. The last part of the discussion felt a little bit too much like a summary. Whilst important points are raised, are there any other published studies that support the notion of divergent CLP effects depending on concentration or timing etc. Could these effects relate to the ability of Chil1 to bind IL-13 decoy receptor at high versus low concentrations etc. A more thorough discussion of some of these aspects raised would help raise the profile of this manuscript. Perhaps even including a summary diagram considering the complexity of the findings could help highlight key points from the data?

Minor comments

Generally, the use of Chil1, Chil3, Chil4 is reserved for describing the gene, and BRP39, Ym1, Ym2 used to describe the protein. I found this a little confusing especially for the western blot data, as I was almost expecting this to be PCR data.

It was unclear why figure 1a was a box & whiskers plot where as 1b was a bar chart. There needs to be some consistency here or an explanation as to why?

Subsection titles in the results could be more descriptive e.g. "Chil1 proteins against further phenotypes" isn't very descriptive as a stand alone section.

It would be useful to show statistics on all graphs rather than just stated in the figure legends.

Some figures show n numbers within the figure, and others not? E.g. Fig 4D-F versus 4a-c?

Reply addressing reviewers comments

(new text is highlighted in the manuscript in yellow. Text referred to in reply is highlighted in the manuscript in green)

Reviewer #1

Reviewer 1 makes multiple very positive comments about the manuscript, stating it to be a “thorough” and “comprehensive” investigation and noting several strengths of the manuscript. Points to be addressed:

1. The study relies heavily on mouse models, and the translational relevance to human cancers is not directly addressed or discussed.

Reply

Additional text is now included in the discussion to highlight the relevance to humans:

Human and mouse CHI3L1 share 73% amino acid identity and are structurally highly similar. In humans, CHI3L1 expression is upregulated in the tissues of numerous inflammatory disorders and multiple cancers, including the EBV-associated cancer nasopharyngeal carcinoma (NPC). Overexpression of CHI3L1 in NPC cells is linked with tumour-associated inflammation and proliferation and for this cancer, like several others, CHI3L1 is being considered not only as a diagnostic marker, but as a potential therapeutic target.

And

With tumours that show high level Chil1 expression, Chil1 inhibition may well counteract its tumorigenic properties and be therapeutic, however duration of such treatment would need to be carefully evaluated to avoid loss of any protection Chil1 affords. For tumours with low or physiological levels of Chil1 expression, anti-Chil1 therapy has the potential to be deleterious.

2. The mechanisms by which Chil1 exerts its effects are not fully elucidated at the molecular level. The study does not delve deeply into the signaling pathways affected by Chil1, which could provide more mechanistic insight into its functions. Additional mechanistic studies exploring the differences in Chil1 signaling based on the level of Chil1 expression would be a significant help to address this issue.

Reply

In the introduction we refer to signalling pathways impacted by CHI3L1/Chil1, but cannot review the full panoply of the molecular effects in this text, however we now cite a recent review which provides comprehensive coverage. It would be a very relevant next step to explore the change in signalling patterns when CHI3L1 is expressed at low, versus high levels, but is beyond the remit of the current study.

3. The impact of Chil1 on specific immune cell population and their function is not thoroughly explored.

Reply

In this study, we have only touched on the potential impact on the immune cell population. To do such a study justice would be a significant undertaking and beyond the objectives of the study reported here. However, hopefully this report now

provides the rationale for immunology groups who are well placed for such investigations to explore the issue.

4. The potential interactions between Chil1 and other inflammatory mediators or oncogenic pathways are not extensively investigated.

CHI3L1/Chil1 (like LMP1) has a wide impact on several pathways, and multiple inflammatory mediators are deregulated in the inflamed tissues in the mouse model. We have previously examined and published this (now cited in the introduction) in Hannigan et al 2011. It is likely that the contribution that CHI3L1/Chil1 makes to this complex milieu of factors will take extensive investigation to unravel, but well worth exploring as a future study. In this study, we aimed to define the functional consequences of CHI3L1/Chil1 overexpression (versus absence) and its role in tumorigenesis; to take what has been correlative in many studies to functional consequence. In several reports, the observed CHI3L1/Chil1 overexpression seen in many tumours and disorders has been inferred as evidence for a tumourigenic action. We aimed to examine if the overexpression was causative or consequential in the pathology and hence focussed on the phenotypic results. This can now act as a foundation for further studies examining the affected immune cells and inflammatory factors.

Reviewer #2

Reviewer 2 summarises the data and indicates that it provides valuable information. The reviewer concludes that “Overall, the data would suggest that therapeutically targeting Chil1 in a tumor setting could lead to enhanced tumor growth” but the situation is rather more complex and the reviewer asks for greater clarity. To help with the conclusion, we have added the text in the discussion:

With tumours that show high level Chil1 expression, Chil1 inhibition may well counteract its tumourigenic properties and be therapeutic, however duration of such treatment would need to be carefully evaluated to avoid loss of any protection Chil1 affords. For tumours with low or physiological levels of Chil1 expression, anti-Chil1 therapy has the potential to be deleterious.

Reviewer comment re figure 6 data: For the cell injection experiments (shown in figure 6), the reviewer correctly notes we have used LMP1 transgenic males and WT females as recipients. This is because the transgene for mouse line L2LMP1.117 resides on the Y chromosome (as detailed in M&M), so we have no alternative (although we do have female cell lines derived from an LMP1 mouse line with an autosomal insertion, but this mouse line only exists now in frozen embryo storage). However, the data shown in figure 6D, E, F, G show that potential differences in male/female host responses are unlikely, as there is little or no difference between the sexes of the recipients, primarily the Chil1 genotype matters (which is shown for both sex recipients).

In A to C (using LMP1tg/Chil1WT cells), there is a difference between the LMP1 transgenic status (and therefore sexes) only for the Chil1-WT recipients, while Chil1-KO recipients show the same response in both sexes (lack of tumour growth). This difference could have been due to the sex of the recipients or LMP1tg status of the recipients, but our further experiment (shown in D) supports the latter. For example, females might show a significantly better tumour cell rejection (or poorer tumour cell

growth) than males. Conversely, the LMP1tg injected cells might be better rejected in the WT mice, responding to LMP1 as a foreign antigen.

To distinguish between these possibilities, we used wild type cells in the injection (no LMP1 antigen, shown in panel D). These cells produced no difference between the LMP1tg/Chil1WT-male and WT-female recipients. This supports the idea that we are not observing a male/female response in these experiments, but responses that are dependent on the LMP1tg and Chil1 genetic status.

We appreciate this is convoluted and not at all easy to explain.

To help clarify, we have added the text:

The data do not support that a sex difference between recipients is responsible for the difference in tumour growth observed between LMP1tg/male and WT/female recipients (shown in figure 6A, B and C) as this is not seen using WT cells (figure 6D and supplementary figure S9).

Further, with respect to the cell lines, both male and female cells elicited the same response in recipients, which we have noted with the text:

The sex of the injected cell lines did not impact this result.

The reviewer notes that using several cell lines in the injection experiments complicates matters and that using a single cell line would be more straight forward.

We agree with this in principle, however, using any one of these cell lines alone could have led to an incorrect or incomplete understanding. For example, using the WT cells alone (figure 6D), we might have interpreted that Chil1KO recipients cannot support tumour growth, etc. While every tumour cell line will have differences, we do see consistency between cells of the same LMP1/Chil1 genetic status (eg A, B and C in figure 6, E and F in figure 6). Studies are often criticised for using only one cell line, as the finding might be unique to that cell line. We aimed to avoid this issue.

We have added in the text:

In order to gain a comprehensive understanding of the impact of LMP1 and Chil1, we used several different cell lines

The reviewer has questions about the statistical analyses relating to figures 2, and 3&4:

Figure 2 – why T test vs Anova.

For the 44 and 211 day blots, only LMP1tg and LMP1tg/Chil1KO could be statistically compared as there were ≥ 3 samples. For comparing two groups, a T test is appropriate. For the 110 day blot, $n=3$ for all 4 genotypes, so for this, Anova is appropriate where 2 factors are analysed across 4 groups. Note, this panel has now been replaced and no significant difference was found for the factor Chil1, while high significant difference is observed for LMP1. This is now clarified in the figure legend and on the image.

Statistical significance was determined by Anova (110 days), showing high significant difference for LMP1 and no significance for Chil1 across the 4 groups, or student T test examining the impact of Chil1 in the LMP1tg background (44 and 211 days).

Figure 3 and 4 – display of stat sig on graphs. – The data in figure 3, relating to a graphical curve, was analysed for significance by a log rank test. As the curves were statistically analysed (and not a single data point), the statistical information cannot

be presented in the same way as the bar charts in figure 2. In figure 4A to C, again curves were analysed by log rank test. In figure 4D to L, each time point was analysed, by T test (for 2 groups) or Anova (for 4 groups). This would be immensely confusing to try to add the statistical information for every time point on each graph, so this was provided in the legend where possible, or in detail in the supplementary information.

Reviewer #3

Reviewer 3 states that the manuscript offers new evidence and indicates the experiments are well designed and statistically analysed. The reviewer comments that the data are complex and clarification in places would help, perhaps with use of schematics.

We have gone through the manuscript and tried to clarify text in places. We have clarified in the legend, the schematic used in figure 4.

Specific points as numbered by the reviewer:

1] *Grammatical errors and better citation to figure sections.*

We have corrected detected errors and cited the figure parts indicated (eg 1A, B, C and 5E, F).

2] As suggested, what was supplementary figure 1 has been added to figure 1A.

However, we have kept the supplementary figure 1 to provide more phenotypic detail in the text.

3] Figure 2 The graphs show normalised data of the blot images shown and this is now clarified in the legend. We did not have a loading control for the previous 100d sample set and have now repeated the blot, with new samples, taken at 110days old. Due to recent problems with the actin antibody, the data are normalised against total protein for each sample in this new data. Chil4 migrates on an SDS-PAGE gel slightly faster than Chil3 and this is now stated in the figure legend.

4] *Reduction of Chil3/4 on the LMP1tg/Chil1KO mice compared to LMP1tg.*

We have commented in the discussion that the overexpression (compared to WT) of Chil3/4 in the LMP1tg/Chil1KO older mice compared to the LMP1tg is less marked and this could contribute to the phenotype. It is possible that this contributes to the differences seen at later stages, but it still results from Chil1 KO and therefore could be one of the many consequences of knock out of this gene. As such, we can't really comment on the contribution specifically arising from less induced Chil3/4 in older mice.

With respect to the eyeball images in figure 3.

We have indicated in the figure legend that the eyeball images shown in D show the pair of eyes from affected individuals, showing the closed and open eye from the same individual.

The reviewer asks how the data support the claims that Chil1 protects against microphthalmia?

This is shown in figure 3B and C where incidence of periocular inflammation and microphthalmia are significantly more frequent in the LMP1tg/Chil1KO mice compared to LMP1tg mice. As such, WT Chil1 in the LMP1tg mice must be protecting against this phenotype.

5] *Clarification of CC regimens for the data in figure 4*

A schematic for the 3 different regimens is supplied under graphs D, E, F and this is now clarified in the figure legend. A statistical analysis has been conducted for each time point for graphs D to L and for the four groups. Trying to add this information to

the images would hugely confuse plots (which are already information dense) and as such, this has been provided in the supplementary information.

Re: figure 6

We are unclear what kind of schematic could help to explain the data in figure 6. The tumour cells were injected once and tumours allowed to form, at which point they were collected and weighed. We have added further text in the legend and main text to try to clarify.

Text: Cells were transplanted **by a single** subcutaneous injection into recipients of the 4 genotypes and tumour development monitored. **The time to tumour development varies for each cell line, consistent with its growth characteristics, however for each cell line, tumours were collected from all mice in the cohort at the same time, to allow comparison.**

Legend: **Tumours were allowed to form for a defined period specific to each cell type, and were then collected and weighed.**

6] Text has been added to the discussion (as indicated under response to reviewer 2) to emphasize relevance to the human condition. We are not aware of other studies that have related CLP levels to divergent properties, other than the many publications (as cited in the introduction) that CHI3L1 levels are high in multiple cancers and inflammatory disorders.

Minor comments

Terminology of the genes and proteins

We agree, the terminology can be confusing and there are several alternative names, with the approved names having changed in the last years.

The accepted gene and protein name according to the HGNC (human gene nomenclature committee) for CHI3L1 is CHI3L1 (aka GP39, YKL40, YK-40, CG-39). However, the inclusion of the "3" in the name is not entirely logical and does lead to confusion. Nevertheless, we have used the recommended name for the human gene/protein in this manuscript.

The accepted gene and protein name for the mouse orthologue, according to the MGI mouse nomenclature committee is Chi3l1 (aka Chil1, Brp39, Gp39). In order to avoid confusion with Chil3 (and to shorten the KO name just a little bit), we have used the alternative name in our study of Chil1 and clarified this point in the first lines of the introduction and title.

The accepted gene and protein names for the mouse paralogues, according to the MGI mouse nomenclature committee are Chil3 (aka Chi3l3, Ym1) and Chil4 (aka Chi3l4, Ym2).

Given the accepted gene/protein names for Chil3 and Chil4, which are examined in our manuscript, we felt it was less confusing to use the mouse name Chil1, rather than the less logical Chi3l1. To use a different name for the protein compared to the gene, we feel would just confuse matters further.

Re data shown in what was figure 1a and 1b, now 1b and 1c - these are different types of data, the first (shown as box plots) relate to a single point in time for each mouse (the age of transition from one stage to another). The second set of graphs show a duration of time, better represented as bar charts. To help with this difference, we have amended the Y axis titles in the revised figure.

Sub section titles - we have tried to give informative titles without making them overly long.

Statistics depicted on some of the figures (such as figure 4) would overly complicate the graphs and make visualisation of the data harder. Where visualisation of the statistics does not obscure the data, we have shown these on the figure.

The n numbers shown in figure 4D-F apply to all of the graphs for that regimen. This is now clarified in the legend.

Number of mice (n) entered on to each of the three regimens (applying to all graphs for that regimen) is indicated in D, E, F

July 9, 2025

RE: Life Science Alliance Manuscript #LSA-2024-03138-TR

Prof. Joanna Beatrice Wilson
University of Glasgow
School of Molecular Biosciences, College of Medical, Veterinary and Life Sciences
University of Glasgow
Davidson Building
Glasgow, G12 8QQ G12 8QQ
United Kingdom

Dear Dr. Wilson,

Thank you for submitting your revised manuscript entitled "Upregulated chitinase-like protein1 promotes tumour growth while physiological levels are protective". Reviewers are satisfied with the changes in place. We would be happy to publish your paper in Life Science Alliance pending final revisions necessary to meet our formatting guidelines.

- Please be sure that the authorship listing and order is correct.
- Please upload all figure files as individual ones, including the supplementary figure files; all figure legends should only appear in the main manuscript file.
- Please add the X and Bluesky handles of your host institute/organization as well as your own or/and one of the authors in our system.
- The titles in the system and the manuscript file must be consistent.
- Please be sure that the author order is correct and matches between the system and the manuscript file.
- Please add your main and supplementary figure legends to the main manuscript text after the references section.
- We encourage you to revise the figure legend for Figure S7 such that the figure panels are introduced in alphabetical order.
- There is a call-out for figure 1D, and this figure doesn't have panel D. Please correct.
- Please be sure to add call-outs for panels A and B for supplementary figures S3-S10.
- Please add molecular weight markers to all protein blots.
- Please modify Figure 5E by adding a line or bar or separate the images to clearly indicate that the last lane was run on a separate gel.

A. FINAL FILES:

-- Summary blurb (enter in submission system): A short text summarizing in a single sentence the study (max. 200 characters including spaces). This text is used in conjunction with the titles of papers, hence should be informative and complementary to the title. It should describe the context and significance of the findings for a general readership; it should be written in the

present tense and refer to the work in the third person. Author names should not be mentioned.

B. MANUSCRIPT ORGANIZATION AND FORMATTING:

Sincerely,

Reviewer #1 (Comments to the Authors (Required)):

The revision is relevant and sufficiently addresses the translational concern. The human cancer connection is specifically highlighted with examples (NPC), and the therapeutic implication is discussed with nuance. A minor improvement could have included more detailed mention of existing anti-CHI3L1 strategies (e.g., antibodies, small molecules), but this is not essential

Reviewer #2 (Comments to the Authors (Required)):

This revised manuscript has largely addressed the concerns of reviewers. in particular clarifying some complex experimental set-ups and thus interpretations.
While additional mechanistic work would have been valuable, it is understandable that this would be a significant extension of what is already a considerable amount of work.

Reviewer #3 (Comments to the Authors (Required)):

The authors looked to have addressed all the reviewers concerns to a satisfactory level. Some points would have been good to address, but I do understand this would be beyond the scope of this publication

July 30, 2025

RE: Life Science Alliance Manuscript #LSA-2024-03138-TRR

Prof. Joanna Beatrice Wilson
University of Glasgow
School of Molecular Biosciences, College of Medical, Veterinary and Life Sciences
University of Glasgow
Davidson Building
Glasgow, G12 8QQ G12 8QQ
United Kingdom

Dear Dr. Wilson,

Thank you for submitting your Research Article entitled "Upregulated chitinase-like protein1 promotes tumour growth while physiological levels are protective". It is a pleasure to let you know that your manuscript is now accepted for publication in Life Science Alliance. Congratulations on this interesting work.

DISTRIBUTION OF MATERIALS:

Again, congratulations on a very nice paper. I hope you found the review process to be constructive and are pleased with how the manuscript was handled editorially. We look forward to future exciting submissions from your lab.

Sincerely,
